# A Comparative Study of the Tourism Carrying Capacity of the State of Baja California between 2019 and 2022

**Blanca Bernal *** , **Nancy Montero and Sergio Vázquez**

School of Accounting and Administration, Autonomous University of Baja California, Tijuana 22390, Mexico; nmontero@uac.edu.mx (N.M.); sergio.vazquez@uabc.edu.mx (S.V.)
* Correspondence: blancab@uabc.edu.mx

**Abstract:** The purpose of this work is to conduct a comparative study of variations in the indicators of the tourism carrying capacity in the state of Baja California. It is crucial to consider that the state had to confront the COVID-19 pandemic, during which tourism was not deemed an essential activity. This circumstance generated numerous social, psychological, and economic effects, primarily. In this regard, the aim is to identify the consequences of organizing events that promote tourism, particularly concerning the opinions of business professionals in the region. This is a qualitative and longitudinal study; the initial phase took place in May 2019, while the second survey occurred in the summer of 2022. The statistical sample is non-probabilistic and based on convenience, comprising 320 tourism businesses. The findings indicate that the tourist destinations remained appealing, experiencing inflows just above the average and approaching their capacity limits. Significantly, there are areas for improvement in terms of their tourist load capacities across each of the dimensions studied, despite the global health crisis.

**Keywords:** sustainable tourism; sustainable development; carrying capacity; competitiveness; COVID-19 pandemic

## 1. Introduction

The health crisis caused by COVID-19 has generated uncertainty in the global economy and all sectors. Particularly, start-ups and micro-, small, and medium enterprises (MSMEs), especially those in the tourism sector, have suffered significant consequences. The manifestation of a pandemic in the year 2020 was unexpected news. Although it is true that world economic development did not show constant growth, the entire world and its governments were not prepared to face this situation, impacting health, social, and economic issues significantly. The pandemic was declared on 11 March 2020 by the World Health Organization [1]. This declaration had a clear impact on competitiveness, as highlighted in the World Economic Forum report (2019) [2]. Mexico obtained 64.95 points on a scale from 0 to 100 on the Competitiveness Index. This index measures how a country utilizes its resources and its capacity to provide its inhabitants with a high level of prosperity [3]. In 2021, it dropped to 48.6 points, according to the International Institute for Management Development [4].

In the 2023 edition, the Competitiveness Index reveals that Baja California Sur has climbed two positions compared to 2022, securing the fifth position, and Baja California Norte has ascended three positions, reaching the tenth spot. This reflects that 48% of foreign investment is concentrated in the state. However, concerning the perception of the security and crime incidence, the state finds itself in the last positions of the Law sub-index, with an average security perception of 13% and a crime incidence of 24.7%. In contrast, in the Society sub-index, which gauges the quality of life of inhabitants based on inclusion, education, and health conditions, there is a three-percentage-point increase, positioning the state at number five. Meanwhile, in the Governments sub-index, which assesses how

state governments positively influence the competitiveness of their respective states, only a one-percentage-point improvement is observed [5].

When discussing sustainable tourism, the primary objective is to minimize the negative impact on the communities in which it is developed while maximizing the benefits for all involved. This involves promoting local economic activity and respecting the environmental impact and cultural traditions as much as possible. Hence, in tourism planning and management, the impact of the activity on the local population is a constant concern directly related to sustainable development, encompassing economic, social, and environmental components [6].

As is well known, entrepreneurship is crucial to the global economy, and it is estimated that MSMEs account for 90% of enterprises worldwide (99.8% in the European Union and 99.9% in the United States). They provide 70% of the total employment globally and are significant contributors to GDPs, comprising around 50% in the Organization for Economic Cooperation and Development (OECD) countries and 40% in emerging economies [7].

The new environment for small and medium-sized enterprises (SMEs) in Mexico has accelerated digital transformation, leading to the increased adoption of e-commerce. Digitizing a company poses a challenge due to the various changes involved in modifying the business model. However, this growth also brings forth a series of challenges. Notably, competitiveness and resistance to change are commonly referenced issues in MSMEs, especially when it comes to adopting new technologies or sustainable practices. Companies have been compelled to integrate terms like innovation and systematization into their administrative, financial, production, and marketing processes due to the confinement imposed by the COVID-19 pandemic. Additionally, another challenge is the scarcity of financial resources and trained personnel, which can impede the implementation of environmental measures [8].

The objective of this study is to conduct a comparative analysis of variations in the tourist carrying capacity indicators within the tourism industry in the state of Baja California, building upon Bernal's work [9]. This comparison spans two stages: an initial assessment and a subsequent evaluation after the COVID-19 impact in 2022. The significance of this study lies in determining whether the tourism offerings were planned based on the recommendations outlined in the preliminary study. The aim is to ensure that these offerings do not cause physical damage to the natural or artificial environment of the region and that they avoid emotional, economic, and social harm to the local community and culture. It also aims to strike the right balance between development and the conservation of cultural traditions. Additionally, for entrepreneurs, understanding the tourism carrying capacity is essential for managing their businesses sustainably and ensuring the preservation of the natural and cultural resources in their operational areas.

## 2. Literature Review

The Secretary of Tourism of the Government of the State of Mexico [10] notes that the COVID-19 pandemic has resulted in a significant paralysis of tourism activities in Mexico. Travel restrictions, border closures, and social distancing measures implemented to contain the spread of the virus directly impact tourism micro- and small- and medium-sized enterprises (MSMEs) that heavily rely on the influx of visitors. In the initial phases of the pandemic, government restrictions and public health concerns led to the temporary closure of many tourism businesses, affecting MSMEs such as hotels, restaurants, travel agencies, and local tourism services. Simultaneously, the "Think of Mexico, Think of…" campaign was implemented as part of the digital strategy to mitigate the crisis caused by COVID-19 and support Mexico's tourism sector. The goal was to ensure that, despite the confinement situation, Mexico would remain in the minds of potential travelers from the main issuing markets.

The State Performance Evaluation System highlights a lack of facilities or easiness for MSMEs and entrepreneurs in the state of Baja California to access financing, training, and support for initiating and/or strengthening their businesses. This, in turn, affects

the improvement in family economies and the reduction in unemployment. Additionally, the negative effects of COVID-19 on the state's economy exacerbated the challenges. The population of Baja California encountered increased difficulties in obtaining and retaining employment, resulting in a decrease in the employment rate from 4.87% in 2020 to 2.3% in 2021. Inflation rose from 4.41% in 2020 to 6.08% in 2021, reducing the purchasing power of family economies, entrepreneurs, and/or MSMEs, placing Baja California at the 12th lowest level in the country. The national average was 2.93% [11].

The state of Baja California is considered a significant job generator in Mexico, owing to its close proximity to the United States of America. This proximity leads to a consistent flow of temporary migrants initially, and, after a brief period, many decide to stay and establish permanent residence in the state. The year 2020 posed exceptional challenges for the global economy, and Baja California was no exception. During this difficult and atypical year, the state experienced a substantial economic contraction, resulting in a loss of 96,143 formal jobs in the first half of the year, representing a 5.11% decline in employment. Additionally, Baja California witnessed an economic contraction of 19.2% in the State Economic Activity ITAEE during the second quarter, compared to the national indicator of 17.3%. This indicates that the personal finances of families in the state were severely affected [12].

It should be noted that Mexican public policy does not actively encourage private investment, and the country lacks anti-cyclical fiscal adjustment policies in the productive sector. This not only impacts the short-term recovery of productive activities but also hampers long-term prospects. From July 2018 to March 2020, foreign investment in Mexico decreased by 16%, and the total amount of foreign investment continued to decline, reaching a reduction of up to 32% in April and May. These contractions were a result of tightened financing channels for Mexican MSMEs [13]. Nevertheless, one out of every ten surviving MSME businesses received some type of financing between May 2019 and July 2021, with 75.2% using it to purchase supplies [14].

The tourism carrying capacity refers to the maximum number of visitors that a destination can support without compromising its environmental, cultural, or social integrity [15]; it serves as a tool to measure indicators of sustainable tourism, involving the careful planning of how many visitors a region or tourist destination can accommodate without exceeding its limits to the point of irreversible damage [16]. In assessing the tourism carrying capacity, it is essential to consider the impacts caused by visitors, as well as the perceptions and expectations of the study participants. The primary objective of determining the tourist carrying capacity is to identify when congestion and bottlenecks in tourism development may begin.

The evaluation of the carrying capacity applied to the management of tourism sites can be based on quantitative methods (calculating numerical standards for quantification) and qualitative methods (proposing management models for protected areas). The literature provides several models for studying sustainable tourism. Several of the primary models used to measure the tourism carrying capacity are listed below (see Table 1) [7,15–21].

**Table 1.** Comparative table of main models for measuring tourism carrying capacity.

| Model | Authors | Approach/Main Characteristics |
|---|---|---|
| Cifuentes and Salinas | Cifuentes, L. A., and Salinas, E. R. | Considers environmental and social factors. Focuses on the relationship between natural resources and tourism activities. |
| Christaller | Walter Christaller (1933) | Central Place Theory analyzes the spatial distribution of tourist attractions in relation to the carrying capacity. |
| Visitor Day Equivalent (VDE) | George C. Coggins (1969) | Measures the carrying capacity based on visitors and their lengths of stay, providing a standardized unit for comparisons. |
| Gunn | C. A. Gunn (1972) | Introduces the concept of an "acceptable carrying capacity", evaluating the relationship between visitors and resources, and considering the quality of the visitor experience. |

**Table 1.** *Cont.*

| Model | Authors | Approach/Main Characteristics |
| --- | --- | --- |
| Limits of Acceptable Change (LAC) | Stankey, G. H., Cole, D. N., et al. (1982) | Management approach establishing acceptable limits of change in a tourist area and guiding actions to maintain environmental quality. |
| Mathieson and Wall | Mathieson, A., and Wall, G. (1982) | Tourism planning model incorporating carrying capacity and its relationship with economic, environmental, and social impacts of tourism. |
| Mill and Morrison | Mill, R. C., and Morrison, A. M. (1985) | "Carrying Capacity Theory" approach analyzing limitations and opportunities for sustainable tourism development in a specific area. |
| Pressure–State–Response Model | World Commission on Environment and Development of the United Nations (1989) | The main idea is that human activities (pressures) impact the state of the environment, and political and management responses are necessary to achieve sustainable development. |
| Porter | Michael E. Porter (1990) | Competitive approach to analyze carrying capacity from the perspective of a destination's competitive advantage. |
| Crouch and Ritchie | Crouch, G. I., and Ritchie, J. R. (1999) | Sustainable tourism demand model considering the relationship between tourism supply and demand, focusing on sustainability. |
| Decision Support System (DSS) | Weaver, D. B., and Lawton, L. J. (2001) | Utilizes a decision-making approach to manage tourism carrying capacity, considering economic, environmental, and social factors. |

It is important to note that these models may have specific applications and advantages in particular contexts. Additionally, ongoing research in the field of tourism may lead to the evolution and development of new approaches and models.

For this paper, we employed the Pressure-State-Response model, which posits that human activities exert pressure on the environment, leading to observable changes. Therefore, maintaining ecological balance becomes crucial. The model is fundamentally grounded in the logic of causality, presupposing relationships of action and responses between economic and environmental activities. This model prompts a series of straightforward questions: What is affecting the environment? What is the current state of the environment? What measures are being taken to mitigate or resolve environmental problems? Each of these questions is addressed through an established system of indicators [17].

After reviewing the existing models, the Pressure–State–Response model was chosen because it provides a systematic approach to understanding the interrelationships between human actions, the state of the environment, and social and political responses. The model is based on the idea that human activities generate pressures on the environment, affecting its state and conditions. "Pressures" are human actions such as pollution and resource extraction that negatively impact ecosystems. "State" refers to the environmental quality, encompassing aspects such as the air and water quality, biological diversity, and ecosystem health. "Response" involves actions and policies implemented by a society to address pressures and improve the state of the environment, including environmental policies, regulatory measures, behavioral changes, and sustainable technologies.

Similarly, the PER model facilitates the creation of specific indicators to measure pressures, states, and responses, providing a quantitative basis for evaluation. It also aids in designing environmental management policies and strategies by providing information on how human actions affect the environment and how these impacts can be effectively addressed.

It should be highlighted that the tourist carrying capacity is a concept predominantly discussed in the academic literature, and primarily from a theoretical perspective. There are few empirical studies related to this concept, although there has been a growing trend in recent years. This is mainly viewed as a tool to underscore the significance of strategies established for the sustainability of companies, aligning actions to mitigate the ecological crisis through business development planning in tourism. This approach enables the development of strategic plans [18]. Based on the definition of the load capacity [9,19], six dimensions are proposed to assess the balance of any tourist destination in compar-

ison to the tourist carrying capacity: the ecological, urban planning, cultural, economic, institutional, and resident psychological capacity dimensions (see Table 2).

**Table 2.** Tourist carrying capacity dimensions.

| | |
|---|---|
| Ecological CC | The maximum level of tourism that allows for preserving the balance of the natural environment in a tourist destination. |
| Urban Planning CC | The maximum level of tourism that allows for preserving the balance of the urban environment in a tourist destination, mainly urban infrastructure and equipment. |
| Cultural CC | The maximum level of tourism that allows for preserving the balance of the cultural environment in a tourist destination, essentially traditions and manners, as well the historical–artistic legacy. |
| Economic CC | The maximum level of tourism that allows for preserving the balance of the economic environment in a tourist destination, making compatible the economic capacity of the receiving community and the economic benefits provided by the tourism in the locality. |
| Institutional CC | The maximum level of tourism that allows for preserving the balance of the political environment in a tourist destination, making compatible the efforts of public administrations to regulate and control tourism growth and citizen service. |
| Resident Psychological CC | The maximum level of tolerance of the residents in a tourist destination, based on querying the balance of the psycho-social environment of a tourist destination. |

Based on the aforementioned, the Pressure–State–Response model was chosen to assess the optimal sustainability indicators for evaluating the tourist destinations. This evaluation considered the present conditions of the municipalities in Baja California from the institutional, economic, urban, ecological, and psychological perspectives. Information pertaining to the circumstances in which cultural and festive events are presented in each tourist destination was considered. With regard to this, the authors of [19] assert that such practices directly contribute to the triple balance in achieving sustainable development.

In the international literature, it was observed that the riparian strip of the Chillon River in the Los Olivos district underwent an evaluation of the influence of the Pressure, State, and Response (PER) indicators on sustainable development [22]. The Pressure indicator aimed to identify economic activities in the area, the State indicator provided information on the observed condition of the area, and the Response indicator allowed for the identification of solutions to address the damages identified in the river. The results underscored the significance of the PER indicators in comprehending the activity, causes, and consequences of pollution in the area, thereby providing valuable information to propose appropriate solutions.

At the state level, a study is presented on the tourist carrying capacities (TCCs) of two trails in the Nuevo Centro Poblacional Ejidal (NCPE) in Baja California Sur, an ejidal zone with Environmental Management Units (UMAs) seeking to enter into alternative tourism. The study was based on the estimation of three main variables: (1) the Physical Carrying Capacity (CCF), (2) the Real Carrying Capacity (CCR), and (3) the Effective Carrying Capacity (CCE). The results indicate that there are two trails with tourism potential in the entity, with estimated Effective Carrying Capacities (ELCs) of 18 visitors per day for the shrimp stream trail and 10 visitors per day for the El Saucito stream trail. These figures serve as reference points to establish visitor control measures for both trails [16].

### 3. Materials and Methods

*3.1. Materials*

The survey is structured with closed questions, mostly multiple choice, comprising 70 items distributed across four analysis sections. First, the socioeconomic and demographic data of the respondents and the type of company they belong to are examined. Second, key economic factors are considered, such as the sales behavior, types of clients, individuals responsible for the proportions and event publicity in each tourist destination, and the use of digital platforms. The economic benefits and beneficiaries of this activity are

also addressed. Thirdly, social aspects related to the tourist sustainability of the state are studied, including the degree of satisfaction of the inhabitants with tourists' behavior during events or tourist services, the services guaranteed by the government during massive events (security, transportation, parking lots, restrooms, etc.), and the type of policies that should be implemented on days with higher tourist influxes to maintain security and order. Finally, questions address the measures that should be taken to protect the flora and fauna in the different evaluated tourist destinations (environmental factors).

Regarding the reliability of the instrument, the survey underwent the Cronbach's Alpha reliability or validity test, obtaining an index of 0.806. This implies that the instrument is highly reliable, as it surpasses the established mean [20]. Generally, a score above 0.60 is considered moderately reliable, a score above 0.75 is reliable, and a score above 0.80 is highly reliable.

### 3.2. Method

This is qualitative, descriptive, and longitudinal research that aims to identify variations in the tourist carrying capacity indicators in the state of Baja California during two specific time periods. The first period covers mid-2019, during the peak contagion rates of the global COVID-19 pandemic, and the second covers the end of 2022, when the economy began to reactivate [23].

Assessment Scale

Regarding the assessment scale, Table 3 presents the criteria for evaluating the tourist carrying capacity. The participants' opinions were taken into account in its preparation, and the scores were established in a percentage range from 0 to 100 [24]. The final result is obtained by calculating the scores in relation to the number of participants. This result determines the percentage of general satisfaction for each analyzed indicator of the tourist carrying capacity or for the variables specifically studied.

**Table 3.** Criteria for assessing carrying capacity.

| Value | Limit | % | Rating |
|---|---|---|---|
| 0 | <limit | ≤35 | Dissatisfied |
| 1 | <limit | 36–50 | Unsatisfactory |
| 2 | At the limit | 51–75 | Moderately satisfactory |
| 3 | >limit | 76–89 | Satisfactory |
| 4 | >limit | ≥90 | Very satisfactory |

Therefore, the lower the score of each participant, the lower the final percentage achieved. This indicates potential areas for regional development in Baja California. Meanwhile, higher scores suggest that regional tourism participants have implemented efficient strategies. Likewise, based on the average score, it can be determined whether the locality is at the limit of its tourist carrying capacity or below or above it.

### 3.3. Sample

For the sample calculation, the universe composition was initially determined using the 2019 database of the North American Industrial Classification System from the National Institute of Statistics and Geography [25]. In the state of Baja California, 12,002 microenterprises (0–10 employees), small enterprises (11–50 employees), and medium enterprises (51–100 employees) in the tourism sector were identified by municipality. The distribution is presented in Appendix A. It should be noted that the sample calculation was non-probabilistic and by convenience, based on the criteria of convenience and practicality, instead of calculated

using a random method. The sample was calculated using the formula for finite populations, considering a 90% confidence level and a 10% error (see Table 4):

$$n = \frac{Z^2 Z2 * N * p * q}{e^2 (N-1) + (Z^2 * p * q)} \tag{1}$$

**Table 4.** Data sheet distribution of the stratified sample by municipality.

| Municipality/Department | Population | Sample | Total |
|---|---|---|---|
| Ensenada | 2100 | 66 | |
| Rosarito Beaches | 647 | 62 | |
| Tijuana | 5420 | 67 | |
| Tecate | 365 | 58 | |
| Mexicali | 3490 | 67 | 320 |

As shown in the table above, the samples are very similar among the tourist destinations studied. This similarity is primarily attributed to the homogeneity of the population, the chosen sampling method, and the absence of specific selection criteria, among other factors [26].

Regarding the sustainability model implemented in this study, the Pressure–State–Response (PSR) model was selected. Its premise is causality, presenting the relationship between the pressure exerted by human beings on the social, economic, and environmental factors of any region. Simultaneously, it allows us to rationalize the abuse and deterioration of livelihoods that affect tourism activity [27]. This model establishes six dimensions that must be analyzed to measure the limit of the tourist carrying capacity: the ecological, urban planning, cultural, economic, institutional, and psychological dimensions.

## 4. Results and Discussion

The tourism industry is of paramount importance to the national economy, playing a crucial role in fostering regional economic development, continual job generation, and enhancing the quality of life of its population. Therefore, to achieve the sustainable development of tourism, certain basic conditions must be met, including a sustainable contribution from economic, social, and environmental factors. This study outlines the main opportunity areas concerning each of the tourist carrying capacity indicators, proposes strategies, and highlights the associated benefits. To facilitate a comparative analysis of the results obtained from both periods, these are mainly presented through contingency tables (cross-tabulations). Furthermore, this section is divided into three parts: the first describes the characteristics of the users of the tourist services studied in each period; subsequently, it shows the data related to Economic Unit participants; then, the variations in the capacities for each analyzed dimension are presented [21].

### 4.1. Subjects of Study

Regarding the type of tourist service users studied, it was found that there are three categories: residents, national tourists, and foreign tourists. The sample is presented in Table 5. In all the tourist destinations studied, resident tourists predominated (in all municipalities) as the clients who provide greater profitability to businesses in the tourism industry. To calculate the growth rate between years, the following formula was used:

$$Rate = \frac{(Final\ Period - < initial\ Period)}{Initial\ Period} \tag{2}$$

$$Rate = \frac{(2022 - 2019)}{2019} \tag{3}$$

**Table 5.** Changes in the types of customers from 2019 to 2022.

| | Tijuana | | Mexicali | | Tecate | | Rosarito | | Ensenada | |
|---|---|---|---|---|---|---|---|---|---|---|
| | **2019** | **2022** | **2019** | **2022** | **2019** | **2022** | **2019** | **2022** | **2019** | **2022** |
| Foreign | 28.00 | 18.00 | 12 | 10 | 22 | 36 | 27 | 29 | 18.18 | 22 |
| Rate | −35.714 | | −16.67 | | 63.636 | | 7.400 | | 21.01 | |
| National | 8 | 6 | 25 | 15 | 16 | 6 | 21 | 18 | 30 | 25 |
| Rate | −25 | | −40 | | 38 | | −14 | | −17 | |
| Residents | 64 | 76 | 63 | 75 | 62 | 58 | 52 | 53 | 52 | 53 |
| Rate | 31.25 | | 19.04 | | −32.25 | | 1.92 | | 1.92 | |
| Total | 100 | 100 | 100 | 100 | 100 | 100 | 100 | 100 | 100 | 100 |

It was observed that the Tijuana municipality was the most affected in its variation rate, experiencing a decrease of −35.71% compared to the initial year of comparison. This decline can be attributed to the high rate of insecurity in the region, leading to an increase in common crimes and a decrease in the perception of security [5]. However, resident participation increased in all cases.

After identifying the variations in the types of clients requesting tourist services in the five municipalities of the state of Baja California, the variation rate by type of service was analyzed (see Table 6). A greater variation was observed in artistic, cultural, and sports services. It is important to note that the cultural industry was significantly impacted by the COVID-19 pandemic in 2020, with performing arts and concerts being the most affected, experiencing a drop of 9.4%, while the overall economy decreased by 7.9% in the same year [28]. In 2022, the Baja California state government implemented a strategy to support local talent, including artists like Frank Di, a representative of Baja California music, who participated in the Villa del Mar Festival in Chile held in February 2023. Additionally, efforts were made to reinstate cultural spaces and allocate state funds to cultural areas, aiming to bring artistic expressions closer to children and teenagers, promoting their full development and well-being [29].

**Table 6.** Variations in tourist services offered from 2019 to 2022.

| Type of Service | 2019 | 2022 | Total Rate of Change |
|---|---|---|---|
| Temporary housing | 128 | 82.07 | −35.88 |
| Food and beverage preparation | 181.56 | 202.49 | 11.53 |
| Art, culture, and sports | 10.44 | 35.44 | 239.46 |
| Total | 320 | 320 | |

All values are expressed in percentages.

Regarding temporary accommodations, they continued to experience negative numbers (−35.88) in 2022, a consequence of the ongoing health crisis. To address this situation, the state of Baja California designed and implemented the Protocol for the Economic Reactivation of Lodging Services. The primary goal of this protocol is to ensure maximum security for lodging service workers and their entire value chain. It involves adopting appropriate containment, prevention, and control measures to safeguard their health [30].

Company Size

In Mexico, the criteria for micro-, small, and medium enterprises (MSMEs) are mainly based on the number of employees and the industry to which they belong. For instance, companies within the manufacturing industry can have fewer than 501 employees, while in the commerce, service, and agricultural sectors, the limit is up to 250 employees [31].

To maintain consistency with the sample distribution established in 2019, the same criteria were followed in 2022. The distribution of the sample by municipality and size is illustrated in Figure 1. It is evident that microcompanies provided the majority of information for this study. This prevalence is attributed to the fact that most of the companies offering tourist services in the region fall under the category of microbusinesses.

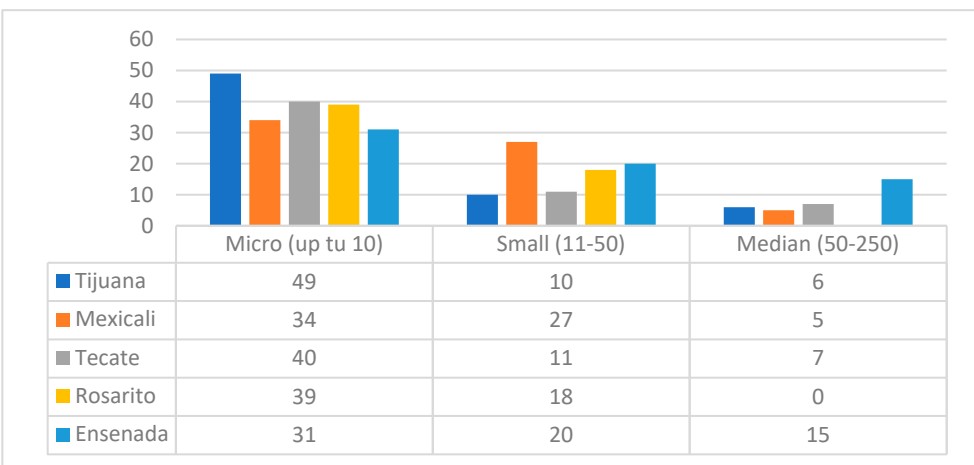

**Figure 1.** Distribution of companies by municipality vs. size.

Micro- and small and medium-sized enterprises (MSMEs) play a crucial and structural role in Mexico's economy. At the beginning of 2020, there were 4.9 million registered MSMEs in Mexico, constituting 78% of private industry employment and contributing to 52% of the total gross production, as reported by Mexico's National Institute of Statistics and Geography [32]. Given their substantial numbers, widespread geographical presence, and significant impact on rural areas and small towns, MSMEs are poised to play a key role in the nation's recovery from the pandemic and in future economic growth [33].

In relation to the average employed personnel of the new ventures, survivors, and closed businesses by state, according to the sector of economic activity and the size of the establishment, it was observed that the proportion of new ventures in Baja California was 20% between the years 2019 and 2022 [31]. Meanwhile, the proportion of closed establishments was 31.67%. These figures are based on the Study on Business Demographics [14].

*4.2. Load Capacity Data*

This section presents the results regarding the carrying capacities of the different tourist destinations studied. This tool analysis is usually applied for a specific place and time. Therefore, the state of Baja California was the setting for this study at two different moments in time (2019 and 2022), generally reflecting the perception of the different entrepreneurs of the micro-, small, and medium enterprises (MSMEs) that participated as providers of some tourist services during cultural events and festivities in each of the municipalities of Baja California. For a better understanding, the carrying capacities are divided into five subsections: Economic CC; Institutional CC; Urban Planning CC; Ecological CC; and Psychological CC.

4.2.1. Economic CC

The sample studied comprised 320 enterprises, distributed by gender and municipality. It was identified that, in all cases, there was greater participation of the male gender, while the participation of women was decreasing (see Figure 2). This is because many women work at home and not in the labor market, with 66 out of every 100 women not working outside the home. Another important factor that demonstrates the decrease in the active participation of women in the labor field is the Economic Participation Rate (EPR), which, in 2021, was composed as follows: out of every 100 men, 72 were economically active, while out of every 100 women, 46 were economically active [34].

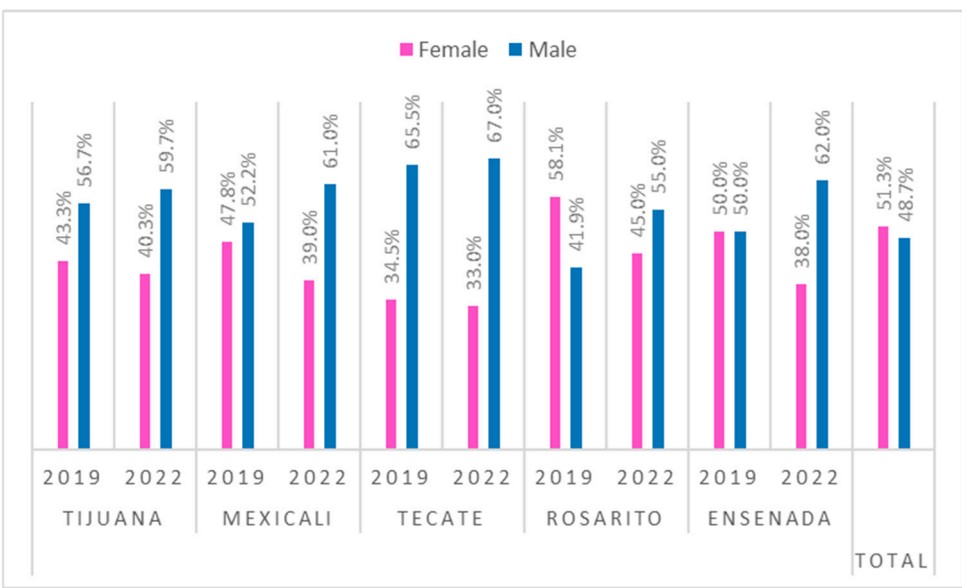

**Figure 2.** Distribution of the sample by municipality and gender. Note: prepared by the authors.

To determine the carrying capacity of the economic component, the economic benefits that local entrepreneurs received to remain competitive and face the crisis caused by the COVID-19 pandemic were analyzed. This involved considering persistent negative and positive aspects, as well as their expectations regarding increased employment. In this regard, positive variations were observed in all the evaluated indicators (see Table 7). However, the negative aspects showed a variation that remains at the limit compared to 2019, implying a risk of financial uncertainty and public safety but, above all, the preservation of jobs.

**Table 7.** Variations in the entity's economic carrying capacities.

| Type of Capacity/Variable | 2019 | 2022 | Variation |
|---|---|---|---|
| Economic CC | | | |
| Benefits | 56 | 92 | 36 |
| Negatives | 51 | 75 | 24 |
| Positives | 75 | 88 | 13 |
| Increased employment | 35 | 91 | 56 |

According to the Economic Panorama of Baja California [35], a comparison between the year 2020 and the first quarter of the year 2023 indicated a 9.8% decrease in working poverty. This statistic proves that Baja California is the third entity in the country that has reduced this indicator the most since COVID-19 [35]. Between January and June 2023, 26,479 new permanent jobs were registered, along with 4716 temporary jobs and 2907 jobs in the countryside, totaling 34,102 formal jobs created. It is noteworthy that, in June 2023, 3000 permanent jobs were created, while 165 temporary jobs were added and 1389 jobs in the countryside were lost. The net balance for the month was 1776 formal jobs. However, by July 2023, the Mexican Social Security Institute reported that the number of new jobs by sex and municipality remains as follows in Table 8.

**Table 8.** Jobs generated by gender and municipality in Baja California.

| Gender | Ensenada | Mexicali | Rosarito | Tecate | Tijuana | Total |
|---|---|---|---|---|---|---|
| Male | 3748 | 4352 | 331 | 782 | 11,152 | 20,365 |
| Female | 2652 | 2764 | 270 | 372 | 7670 | 13,728 |
| Total | 6400 | 7116 | 601 | 1154 | 18,822 | 34,093 |

Note. Based on IMSS database (2021) [36].

As part of the benefits associated with the effective management of the tourist carrying capacity, 34,093 new jobs were created in the state of Baja California. This signifies a remarkable growth rate of 18.19% within a single month, showcasing a significant social impact on the community and its families. This surge in employment demonstrates an increase compared to the situation in 2019. Consequently, Baja California ranks as the seventh state that generates the most and best jobs in Mexico. Notably, over half of the jobs generated in the first semester were concentrated in Tijuana, accounting for 55.2%. Of these jobs, 59.7% were for men and the remaining percentage were for women. The business services industry leads in job creation, contributing to 29.2% of new jobs, followed by construction at 21.2%. This has resulted in a decrease in the unemployment rate for 2023 by 2.42%, positioning Baja California as the twelfth lowest level in the country. The national average for this year was 2.93% [35]. Moreover, an analysis of the unemployed population by age reveals that, among Baja Californians, the highest percentage falls within the age range of 25–44 years, comprising 35.8% [37], with 40.9% women and 33.6% men.

4.2.2. Institutional CC

This indicator reflects the capacity limit at which the city council manages the necessary services, primarily for mass tourism. This section analyzes the security provided, the access to public transportation through different means, the availability of toilets and public parking, as well as the existence of information problems.

Upon analyzing all the indicators in the institutional section (refer to Table 9), a significant improvement is observed between 2019 and 2022. This improvement stems from the strategies implemented by the government of the state of Baja California to maintain optimal levels of the capacity limit. The city council focuses on works that contribute to establishing the entity as a world-class tourist destination. The government achieves this through the creation of creative promotion, dissemination, and public relations strategies. These initiatives strengthen and position the state's image at the national and international levels, resulting in increased visitor numbers, higher average spending, and the generation of more jobs, contributing to social well-being [37].

**Table 9.** Variations in the institutional carrying capacities of the entity.

| Type of Capacity/Variable | 2019 | 2022 | Variation |
|---|---|---|---|
| Institutional CC: capacity limit at which the city council manages the services | | | |
| Security | 75 | 92 | 17 |
| Transport | 15 | 62 | 47 |
| Bathrooms | 8 | 52 | 44 |
| Food stalls | 78 | 91 | 13 |
| Information services | 8 | 82 | 74 |

One of the services that most concerns residents in various tourist destinations is the provision of citizen security. This service serves as a public good, providing assurance to individuals regarding their physical, psychological, patrimonial, and social integrity. It enables the exercise of their rights, such as freedom and peaceful coexistence. Findings on this topic reveal that only 72% of the participants affirmed having these services during their local festivities. However, in 2022, 98% of the sample agreed that there are public and private security services available at all times. In this way, during the business consultation for the formation of public policy for the Baja California government (2021–2040), security was prioritized in second place by up to 62%, as it currently holds the greatest weight in the state economy at this time [37].

As part of the Baja California government's strategies, there is a proposal to establish a thematic axis called mobility (transportation). This initiative aims to create the necessary conditions to comprehensively, sustainably, effectively, safely, and equitably meet the population's mobility needs. It also seeks to promote the metropolitan area as a logistics node, covering projects such as the metropolitan integration of the public transportation

system, sustainable mobility, the strengthening of the metropolitan road system, improvement in and expansion of the highway system, and the strengthening of the cross-border corridor system.

Another important factor in the institutional capacity is the availability of toilets (public bathrooms). In 2019, as shown in Table 9, they were below their carrying capacity limit, leading to significant dissatisfaction. They were not offered in optimal conditions, and there was an imbalance between the numbers of visitors and available toilets during holidays and/or mass events. This directly affected the general quality of the tourist service. Microentrepreneurs from Baja California companies rated them negatively (8). However, in 2022, the situation improved, with greater satisfaction and a positive variation of 44%.

Regarding information services, the government of the state of Baja California [38] designed and implemented a website where prospective visitors can consult various tourist destinations in advance. The platform provides information on activities, scheduled events, and essential travel preparations. Guides on the consulates of other countries, tourist assistance through the emergency number 078, and weather information are also available, among other resources.

### 4.2.3. Urban CC

In the urban carrying capacity indicator, the measurement of the capacity can be achieved, allowing for the determination of the sizes of the tourist gatherings in the settings of each city. This, in turn, enables a greater promotion of events and a better projection of attendees. The application of visitor attention processes and the development of contingency plans are crucial to ensure that the load capacity is not exceeded (see Table 10).

**Table 10.** Variations in the urban carrying capacities of the entity.

| Type of Capacity/Variable | 2019 | 2022 | Variation |
|---|---|---|---|
| Urban CC | | | |
| Restaurants | 75 | 98 | 23 |
| Distances | 15 | 93 | 78 |
| Access routes | 8 | 65 | 57 |
| Places of recreation and leisure | 78 | 89 | 11 |

It is important to highlight that the geographical proximity to the border of the United States of America provides a competitive advantage to the region. It is one of the main sources of tourists worldwide, generating a competitive edge compared to other destinations in Mexico. Nearly 15.8 million international visitors are received annually, with 84% entering Mexico by land, mainly from the states of California, Nevada, and Arizona. The tourism industry of Baja California has significant potential as a development engine, contributing 3.4% to the National Tourism GDP in 2019, adding MXN 51,364 million. It positioned itself in ninth place, with the highest contribution in Mexico and second among the northern border states. Regarding the average number of rooms available and occupied, in 2019, only 46.87% of its maximum capacity was reached, while in 2021, this increased to 51.34% [39].

It is important to note that the region of Tijuana and San Diego was recognized as the World Design Capital 2024, bringing opportunities for local talent, not only in graphic design but also in roads, gastronomy, art, and the commercial area. The main objectives are to demonstrate the effective use of design to promote economic, social, cultural, and environmental development in the chosen region as the capital of world design [40].

In addition to the above, the Tijuana Development Council affirms that for 2024, 46 thousand direct jobs are projected, along with 73 thousand indirect employees. They also estimate 1.4 million nightly visits with lodging reservations to attend various events held in collaboration with the state government, translating into an economic impact of MXN 1.5 billion in the region.

### 4.2.4. Ecological CC

Water management, at 39%, is the most critical environmental criterion that MSMEs prioritize, followed by waste management at 36%. It was determined that MSMEs play a significant role in enhancing the 3R business model—reduce, reuse, recycle [41]. Likewise, it was concluded that MSMEs could enhance their environmental performance, achieve sustainable results, manage waste effectively, and increase their production efficiency through efficient and ecological thinking. This involves reducing waste through the implementation of recycling, reuse, and remanufacturing strategies [42].

Table 11 analyzes the variations in the ecological carrying capacities in Baja California. It is evident that water and energy savings, waste management, sustainable mobility, and responsible consumption are crucial actions for decarbonizing businesses and result in substantial cost savings [43]. Although there was a favorable percentage variation between the years analyzed in this indicator, issues related to air quality and the rationalized use of water remain urgent.

**Table 11.** Variations in the ecological carrying capacities of the entity.

| Type of Capacity/Variable | 2019 | 2022 | Variation |
|---|---|---|---|
| Ecological CC | | | |
| Respect for flora | 75 | 80 | 5 |
| Respect for wildlife | 15 | 76 | 61 |

Air quality is severely impacted by a large number of companies using charcoal or firewood in their production processes, leading to pollution primarily due to smoke emissions. The Baja California government, through the Air Quality Monitoring Network (comprising 14 stations), measures and regulates the consequences of the deteriorating air quality that affects the health of Baja Californians. This deterioration reduces lung capacity and exacerbates broncho-respiratory diseases, resulting in significant economic repercussions, such as increased medical expenses and reduced labor productivity [37]. To address this issue, the state government has undertaken various projects outlined in the Metropolitan Strategic Plan 2034. These projects include ecological and territorial planning, the development of alternative energy sources, comprehensive solid waste management, the comprehensive use of wastewater, risk and vulnerability tracking, the comprehensive management of hydrological basins, the creation of public green areas, and the promotion of environmental education.

Given the substantial concern among business professionals and experts, the concept of a circular economy has emerged. This concept involves reducing the use of raw materials and promoting the reuse of waste, and it is gaining popularity as a pathway toward sustainable growth with green investments over time. The traditional linear model of "extract, make, use, and dispose" is no longer considered viable in the long term [44]. In contrast, the circular economy focuses on designing systems in which products, materials, and resources remain in use for as long as possible.

It is important to highlight that the implementation of an Environmental Management System (EMS) aims to manage environmental aspects, comply with legal requirements, and address risks and opportunities in the environmental context [45]. However, the circular economy model is easier to execute and accelerates the adoption of new practices, allowing for certification that endorses environmental criteria in MSMEs. Among the benefits of implementing an EMS are cost savings, which are achieved through waste reduction and the responsible use of natural resources, such as water, gas, and electricity, to name a few.

### 4.2.5. Psychological CC

Social psychology studies tourism as a social phenomenon. It analyzes how it affects both tourists and the population in a tourist destination. The focus is on studying the social perspective of tourism, tourist motivation, and the perceptions and attitudes of both tourists and residents towards this phenomenon.

Therefore, this section evaluates the perception and importance given by the residents of the tourist destination and the users of such services to psychological factors such as the regional cultural, heritage, natural, historical, and climatic wealth, as well as the volatility in policy. The latter generally has a negative impact on the overall perception of the region and discourages tourism, affecting its economic profile as well [46]. For this reason, the evaluation of business participants in the MSMEs in the state of Baja California is negative both in terms of the perceived negative consequences and the increase in tourism derived from the aforementioned psychological factors (see Table 12).

**Table 12.** Variations in the entity's psychological carrying capacities.

| Type of Capacity/Variable | 2019 | 2022 | Variation |
|---|---|---|---|
| Psychological CC | | | |
| Negative consequences | 75 | 60 | −15 |
| Increase in visitants | 85 | 93 | 8 |

The COVID-19 pandemic has also had a deep psychological impact on tourists worldwide, affecting their behaviors [47]. There is the perceived risk of host communities regarding racial and regional stereotypes. In terms of psychological matters, the reasons for a trip undoubtedly influence the tourist's decision-making process, primarily evaluating the level of relaxation, the region's security, and the protection related to their security, self-esteem, development, and self-realization. This is based, first, on the tourist's own motivation for traveling and, second, on the reason why they want to travel to that particular destination. All of the above are evaluated before making a travel destination decision [46]. Related to risk perception, the participants in this study strongly concluded (95%) that the information channels are not credible and that the majority (78%) of tourists are unaware of the reality regarding the level of crime in the region.

## 5. Conclusions

In general terms, it can be concluded that the current and subsequent government administrations should focus their efforts on the recovery of the tourism industry, given its importance as a job and resource generator. This can be achieved through strategies aligned with the law, accessibility, sustainability, gender equity, and inclusion in order to take advantage of opportunities in the different segments of Baja California tourism. It should be noted that the main competitive advantage of the state of Baja California is its geographical location (bordering the United States). Therefore, the successful management of its carrying capacity is attributed to the strategies implemented by the regional authorities in collaboration with other industry members. The greatest challenge lies in enhancing the safety, image, cost, and value of tourism proposals.

Regarding economic factors, it can be concluded that companies in this sector face multiple challenges to their survival in the current and post-COVID scenarios. In this sense, it is essential not only to focus on internal competencies but also to develop innovation capabilities and business strategies to manage the innovation processes. Despite the crisis caused by COVID-19, tourists continued to arrive in the region, albeit to a lesser extent, mainly to visit their families (53.69%), for medical care (22.6%), and for pleasure only (16.6%), with a total of 12,821,454 tourists reported in the entity.

In terms of the social aspect, there is no doubt that tourists analyze several elements before choosing a tourist destination. The decision-making process is influenced by factors such as age, income, education, health, family size, household size, the residential area in which the tourist destination is located, and the traffic conditions of the region. Additionally, the decision to return to a tourist destination is closely related to the level of satisfaction when purchasing a product or experiencing travel, overshadowing situational influences such as politics, religion, or the perceived insecurity of the country. Therefore, it can be concluded that these behaviors are closely related to creating behavioral loyalty, which is

linked to emotional satisfaction and perceived value. This loyalty is measured through repeat purchases and recommendations to other people.

Previous studies have shown a close correlation between satisfaction and loyalty, which can be attributed to the effective management of the regional image by governments, promoting greater personal involvement and a stronger connection to the place as a tourist destination due to previous personal satisfaction. This implies positive commercial image positioning for the tourist destination.

Finally, concerning environmental care and the constant fight against the effects of climate change, there has been active participation from organized civil society, generating greater awareness and emphasizing the importance of species preservation and environmental sustainability. More and more people, including authorities at all legal levels, academics, and society in general, are getting involved in these efforts.

It was observed that, in the Mexican tourism context, the Pressure–State–Response (PER) model has been used to address issues related to sustainability and tourism management. Studies applying this model typically provide recommendations to ensure that tourism develops sustainably, minimizing negative impacts on the environment and maximizing benefits for local communities. However, no studies were found that propose public policies or business strategies designed with stakeholders, nor did we find in-depth studies conducted on the perspective of the residents in the tourist destinations in this country.

Among the main limitations to conducting this study, the time it took for its completion and the necessary financial resources to conduct the fieldwork stand out. Because it was a state study, expenses for accommodation, food, and transportation were incurred. Another limitation is the complexity of the long-term impact of the gathered information. Despite the mentioned limitations, this study contributes significantly to tourism management in the region. It provides a detailed insight into the current carrying capacity and highlights areas of opportunities and challenges. These contributions can serve as a basis for informed decision making by government authorities, tourism businesses, and other stakeholders aiming to improve the sustainability and efficiency of tourism in Baja California. In summary, despite some inherent limitations, this study provides a solid foundation for future research and offers valuable recommendations for tourism management in Baja California.

For future research, we suggest exploring specific areas that could not be fully addressed in this study due to time or resource constraints. Additional research could delve into the tourist perception of safety, the effectiveness of implemented sustainability measures, and the impact of external factors, such as health crises or natural disasters, on the carrying capacity. Additionally, conducting longitudinal studies to assess the evolution of the carrying capacity over time would be beneficial.

Moreover, it is important to continually evaluate the general perception of tourists, as this is closely linked to user satisfaction. This evaluation helps in designing or adjusting strategies to enhance tourists' experiences in the state. Strategies should be collaboratively designed among local stakeholders, including tourism authorities, local businesses, and community organizations. This collaborative effort ensures cooperation and access to the sampling points of the greatest economic benefit to this economic sector.

Based on the conclusions presented, some recommendations are provided to guide future actions of the government, businesses, and other stakeholders involved in the tourism sector in Baja California: tourism should be revitalized through strategies focusing on legal compliance, accessibility, sustainability, gender equity, and inclusion; the carrying capacity management should be enhanced by collaborating with regional stakeholders for improved safety, image, and value; tourism companies should prioritize internal competencies and innovation, especially amid post-COVID scenarios; targeted awareness campaigns about factors influencing tourist decisions are essential; active participation in environmental initiatives for sustainable tourism development should be encouraged.

**Author Contributions:** B.B. contributed to the design of the study, methodology development, information processing, sector analysis, design and validation of the data collection instrument, and presentation of the results and conclusions. N.M. contributed to the application and validation of

the instrument and the development of the theoretical framework and made final contributions to the discussion of the results. S.V. contributed to the search for previous studies and the updating of the bibliography and its translation into English. All authors have read and agreed to the published version of the manuscript.

**Funding:** This research received no external funding.

**Institutional Review Board Statement:** Not applicable.

**Informed Consent Statement:** Informed consent was obtained from all the subjects participating in the study. During the two periods of data collection, the objective of the study was explained to the subjects, and their consent was requested before starting the survey; consent was also requested for the publication of the findings.

**Data Availability Statement:** Data from the first part of the study performed in 2019 are available in the section "Espacios Research Data" at https://www.revistaespacios.com/a20v41n31/20413125.html, accessed on 14 October 2023.

**Acknowledgments:** We would like to thank the entrepreneurs of the micro- and small and medium-sized enterprises in the state of Baja California for their participation and support in the research process.

**Conflicts of Interest:** The authors declare no conflict of interest.

## Appendix A

**Table A1.** Distribution of the stratified sample by activity and municipality.

| Municipality: Ensenada Activity | Numbers of Employees | | |
|---|---|---|---|
| | 0–10 | 11–50 | 51–100 |
| (7111) Artistic and cultural entertainment companies and groups | 6 | 1 | 0 |
| (7113) Promoters of artistic, cultural, sporting, and similar events | 10 | 1 | 2 |
| (712) Museums, historical sites, zoos, and the like | 10 | 1 | 0 |
| (7211) Hotels, motels, and similar establishments | 117 | 33 | 3 |
| (7213) Boarding houses and lodging houses, and furnished apartments and houses with hotel services | 7 | 1 | 0 |
| (7224) Nightclubs, bars, canteens, and the like | 72 | 15 | 1 |
| (7225) Food and alcoholic/nonalcoholic-beverage preparation services | 1717 | 101 | 2 |
| | 1939 | 153 | 8 |
| TOTAL | | | 2100 |
| Municipality: Mexicali Activity | Numbers of employees | | |
| | 0–10 | 11–50 | 51–100 |
| (7111) Artistic and cultural entertainment companies and groups | 4 | 9 | 0 |
| (7113) Promoters of artistic, cultural, sporting, and similar events | 10 | 5 | 0 |
| (712) Museums, historical sites, zoos, and the like | 6 | 3 | 0 |
| (7211) Hotels, motels, and similar establishments | 73 | 47 | 6 |
| (7213) Boarding houses and lodging houses, and furnished apartments and houses with hotel services | 6 | 2 | 0 |
| (7224) Nightclubs, bars, canteens, and the like | 133 | 27 | 2 |
| (7225) Food and alcoholic/nonalcoholic-beverage preparation services | 2944 | 206 | 7 |
| | 3176 | 299 | 15 |
| TOTAL | | | 3490 |

**Table A1.** *Cont.*

| Municipality: Ensenada Activity | Numbers of Employees 0–10 | 11–50 | 51–100 |
|---|---|---|---|

| Municipality: Rosarito Beaches Activity | Numbers of employees 0–10 | 11–50 | 51–100 |
|---|---|---|---|
| (7111) Artistic and cultural entertainment companies and groups | 0 | 0 | 0 |
| (7113) Promoters of artistic, cultural, sporting, and similar events | 5 | 0 | 0 |
| (712) Museums, historical sites, zoos, and the like | 5 | 0 | 0 |
| (7211) Hotels, motels, and similar establishments | 18 | 9 | 2 |
| (7213) Boarding houses and lodging houses, and furnished apartments and houses with hotel services | 7 | 2 | 0 |
| (7224) Nightclubs, bars, canteens, and the like | 19 | 2 | 0 |
| (7225) Food and alcoholic/nonalcoholic-beverage preparation services | 536 | 42 | 0 |
|  | 590 | 55 | 2 |
| TOTAL | | | 647 |

| Municipality: Tecate Activity | Numbers of employees Micro 0–10 | 11–50 | 51–100 |
|---|---|---|---|
| (7111) Artistic and cultural entertainment companies and groups | 0 | 0 | 0 |
| (7113) Promoters of artistic, cultural, sporting, and similar events | 2 | 1 | 0 |
| (712) Museums, historical sites, zoos, and the like | 3 | 0 | 0 |
| (7211) Hotels, motels, and similar establishments | 17 | 4 | 0 |
| (7213) Boarding houses and lodging houses, and furnished apartments and houses with hotel services | 2 | 0 | 0 |
| (7224) Nightclubs, bars, canteens, and the like | 16 | 2 | 0 |
| (7225) Food and alcoholic/nonalcoholic-beverage preparation services | 299 | 19 | 0 |

| Municipality: Tijuana Activity | Numbers of employees 0–10 | 11–50 | 51–100 |
|---|---|---|---|
| (7111) Artistic and cultural entertainment companies and groups | 4 | 0 | 0 |
|  | 339 | 26 | 0 |
| TOTAL | | | 365 |
| (7113) Promoters of artistic, cultural, sporting, and similar events | 16 | 2 | 1 |
| (712) Museums, historical sites, zoos, and the like | 8 | 0 | 1 |
| (7211) Hotels, motels, and similar establishments | 173 | 63 | 10 |
| (7213) Boarding houses and lodging houses, and furnished apartments and houses with hotel services | 24 | 1 | 0 |
| (7224) Nightclubs, bars, canteens, and the like | 199 | 58 | 5 |
| (7225) Food and alcoholic/nonalcoholic-beverage preparation services | 4421 | 414 | 20 |
|  | 4845 | 538 | 37 |
| TOTAL | | | 5420 |

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
