# Peer review of "A Comparative Study of the Tourism Carrying Capacity of the State of Baja California between 2019 and 2022"

_sustainability, doi:10.3390/su16103938_

Round 1

Reviewer 1 Report

Comments and Suggestions for Authors

The topic chosen for this study, at a general level, seems interesting and relevant. However, many aspects must be reviewed to be published in a journal of international impact such as Sustainability.

1) The structure of the sections is a little difficult to read as well as the paragraph structure within each section. It is recommended to review the work completely to improve reading and facilitate understanding.

2) It is not explained or justified why the chosen model is ideal in this case.

2) It is indicated that the study is “mixed”, which makes me understand that the collection of information will be both qualitative and quantitative, but this is not observed in the data.

3) The definitive questionnaire and the sources of the items used are not presented.

4) A description of the sample of tourism companies collected is not presented, by size, specific activity, etc.

5) The formula used to calculate the sample size for each stratum is not stratified sampling. Therefore, each stratum has almost the same sampling units regardless of the size of the population in each case.

6) It is indicated that the first wave was collected in “mid-2019” when the pandemic was at its peak, however, the data for Mexico indicates that the first case reported in that country appeared on February 20, 2020. Therefore , the initial sample is not clear that it measures what the researchers claim it measures.

7) The English used is quite poor. As well as the use of periods, semicolons, capital letters, etc. There are too many words and phrases in Spanish.

My general opinion is that authors should spend more time improving the different sections of the work. To make decisions that are more justified and supported by academic literature, and, above all, to better think about the managerial contributions of the results obtained.

Comments on the Quality of English Language

7) The English used is quite poor. As well as the use of periods, semicolons, capital letters, etc. There are too many words and phrases in Spanish.

Author Response

Several reviewers commented that the article was interesting, important and relevent.
However there were some problems (minor and major) with the article that needs to be improved in the next draft.

The comments and corrections requested by the reviewers are presented in strict order of appearance of the structure of the document.

To more easily locate the requested changes in the body of the document, we have highlighted them in blue. Below, we share the changes requested by the four reviewers."

NOTE: MDPI's translation service has been requested.

  1. Abstract:

No citations should be referenced in the abstract. More key words should also be included.

Summary citations have been removed

Keywords were added: Competitiveness and Pandemic Covid-19.( From paragraph 8-25)

  1. Introduction:

Several reviewers noted that the introduction was too long and repetitous. The aim of the study and significance of the study should be clearly written at the end of the introduction section.

The introduction was shortened and the importance of the study was added. (Quotations 8 and 9 were deleted as they were repetitive.)

  1. Literature review:

(not literature reviews!) need to be referenced more, especially between lines 26-32. In addition new literature needs to be included that looks at the post-covid time period as well as the effects in other destinations.

Reviewer 2, 3 and 4. This suggestion has already been addressed by incorporating studies that address effects on other destinations and information on Pot-covid.

A comparative table on the main Tourist Carrying Capacity Models was added (See table 1).

Reviewer 4. Some tables could be replaced by text. The same for figure 1.

The figures 1 was removed from the document; as suggested, its content was explained in their respective locations.

  1. Materials and methods:

Reviewer 1. It is indicated that the study is “mixed”, which makes me understand that the collection of information will be both qualitative and quantitative, but this is not observed in the data.

A correction was made throughout the document that it corresponds to a qualitative study (since it does not qualify as a mixed study).

You also need to provide a stronger justification for why you used the selected model?

Reviewer 1. At the end of the literature, after explaining the main existing load carrying capacity models, it is justified why the Pressure - State - Response model was chosen.

Reviewer 1. It is indicated that the first wave was collected in “mid-2019” when the pandemic was at its peak, however, the data for Mexico indicates that the first case reported in that country appeared on February 20, 2020. Therefore, the initial sample is not clear that it measures what the researchers claim it measures.

The clarification was made that although the contagion was higher in mid 2019; it was not until January 9, 2020 that the Coronavirus was recognized in Mexico; quantified and attended by health institutions.

Reviewer 1.  The definitive questionnaire and the sources of the items used are not presented.

Due to the length of the survey and the fact that it is not required by the publisher, it is not included as part of the document; however, the elements that make up each section are detailed (See in the methodology section).

Reviewer 1. A description of the sample of tourism companies collected is not presented, by size, specific activity, etc.

A description of the sample, by size and specific activity, is provided in the methodology section. Annex 1 presents in greater detail the population that makes up each of the municipalities.

Reviewer 1. The formula used to calculate the sample size for each stratum is not stratified sampling. Therefore, each stratum has almost the same sampling units regardless of the size of the population in each case.

The requested adjustment is made

Reviewer 3. There was no description of the collection process or the method of selection of 320 samples.

In the methodology section, it is explained that the obtained sample consisted of 320 study subjects, based on the total population of Micro, Small, and Medium Enterprises (MSEs) in the sector (economic units)

A description of the sample, by size and specific activity, is provided in the methodology section. Annex 1 presents in greater detail the population that makes up each of the municipalities.

  1. Results

Reviewer 4. Some tables could be replaced by text. The same for figure 4.

The figures 4 was removed from the document; as suggested, its content was explained in their respective locations.

  1. Conclusions

Needed to include a section on limitations of the study, managerial contributions and future research possibilities.

The conclusions already include a section on the study's limitations, contributions to management, and possibilities for future research.

Reviewer 4. The conclusion should not contain any references.

The references (citations) in the conclusions section have been removed.

Reviewer 4. The conclusion could be supported by recommendations.

Based on the conclusions presented, some recommendations are provided to guide future actions of the government, businesses, and other stakeholders involved in the tour-ism sector in Baja Californi

  1. Several reviewers

Noted that your paper was difficult to read, English was poor and the paragraph construction needed to be improved:

This suggestion was addressed; external support was requested for the revision of the translation of the document, and the paragraphs were repositioned with a better structure.7.

My suggestion for a change in your title from...'2019 to 2022' TO 'between 2019 and 2022'.

The recommendation to change the title was heeded: from...'2019 to 2022' TO 'between 2019 and 2022'.

7) The English used is quite poor. As well as the use of periods, semicolons, capital letters, etc. Se realizó una revisión a la traducción original; se realizó una revisión del uso adecuado de la sintaxis.

There are too many words and phrases in Spanish.

All the words in Spanish were translated; as it was reviewed in its entirety; most of them were in the references section..

Reviewer 1 and 3. The structure of the sections is a little difficult to read as well as the paragraph structure within each section. It is recommended to review the work completely to improve reading and facilitate understanding.

The document was organized from the general to the particular and went deeper into the models of tourism carrying capacity, as well as the incorporation of studies related to the measurement of tourism carrying capacity in different destinations.

Reviewer 3. General formatting is incorrect; for example, Chapter 4 is followed by Chapter 5 instead of Chapter 4; Line 74, leave a blank space;

The general format was corrected to provide continuity between chapters; regarding adjusting the formula format for lines 206 to 207: it is not clear what type of adjustment is requested.

Rewiewer 3. Authors' contributions: this section does not belong to the main body of the article.

Authors' Contributions was eliminated as a general section of the document, since it was rightly noted that it does not belong to the main body of the article.

Rewiewer 3. Some tables could be replaced by text. The same for figure 4. Done.

Reviewer 4. What is the significance of putting the number of pages next to each reference?

The page numbers were removed from all citations, as requested.

Reviewer 2 Report

Comments and Suggestions for Authors

This article is very important for all employees in tourism, who have felt the consequences of the Covid-19 pandemic.

Key words are not enough and authors need to add more.

Introduction is too long with plenty of redundant information. It is necessary to shorten it.

Literature Reviews is necessary part of article, but authors must add new research, after Covid-19. It is mandatory to refer to research that deals with the same problem in other destinations.

The research sample is impressive and the methods meet the needs of this type of research.

The results are presented well. Tables and graphs are used adequately. Conclusion is good and enough.

References can be expanded with new research after Covid -19.

Author Response

(The authors gave the same response as above.)

Reviewer 3 Report

Comments and Suggestions for Authors

1. The abstract does not require citation, please reconsider

2. The overall format is incorrect, for example, Chapter 4 is followed by Chapter 5 instead of Chapter 4; Line 74, please leave a blank space; Please adjust the formula format for lines 206 to 207

3. No research significance or gaps were found in this study

4. Literature review: Please supplement the definition, application, and research gaps of the Pressure State Response Model

5. What are the main research methods?

6. There was no description of the collection process and screening method for 320 samples.

7. Author Contributions: This section does not belong to the main body of the article

8. What are the research contributions? What are the theoretical and practical implications? What are the research limitations and future recommendations?

Overall, this study has undergone extensive revisions or even rewriting in terms of writing logic, writing techniques, research motivation, research methods, and literature review.

Comments on the Quality of English Language

Moderate editing of the English language required

Author Response

"Please see the attached file."

Reviewer 4 Report

Comments and Suggestions for Authors

Dear authors

Thanks for your work and presentation.

However, few points could be considered during your revision:

  1. The abstract should not contain any references.
  2. Lines 26-32 should be supported by references.
  3. What is the meaning of putting pages number beside each reference?!.
  4. The aim of the study should be addressed at the end of the introduction.
  5. The conclusion should not contain any references.
  6. The conclusion could be supported by recommendations.
  7. Some tables could be replaced by text. The same for figure 4.

Best wishes

Author Response

"Please see the attached file."

Round 2

Reviewer 3 Report

Comments and Suggestions for Authors

Thank you very much for the revisions made by the author. However, I believe that the modifications still do not address or enhance the previous issues, such as the research significance or gaps, research methods, research samples, literature review, practical implications, and so on.

Specifically, regarding the research sample, although the article mentions the composition of these 320 samples, it doesn't explain why these particular samples were chosen for the study. In other words, does the sample possess representativeness? Normality? Additionally, how was the collection and sampling conducted?

Comments on the Quality of English Language

Moderate editing of the English language required

Author Response

(The authors gave the same response as above.)
